# Comparing Water Use Characteristics of *Bromus inermis* and *Medicago sativa* Revegetating Degraded Land in Agro-Pasture Ecotone in North China

**Zhuo Pang, Hengkang Xu, Chao Chen, Guofang Zhang, Xifeng Fan, Juying Wu \* and Haiming Kan \***

Institute of Grassland, Flower and Ecology, Beijing Academy of Agriculture and Forestry Sciences, Bejing 100097, China

\* Correspondence: wujuying@grass-env.com (J.W.); kanhaiming@hotmail.com (H.K.); Tel.: +086-010-51505207 (H.K.)

**Abstract:** Revegetation with herbaceous plants has been effective in neutralizing land degradation; however, there is limited understanding about the water use characteristics and influences on soil water dynamics of revegetated species for ecological restoration. Hence, the stable isotopic composition of xylem water, soil water and groundwater was measured to investigate the water uptake patterns of *Bromus inermis* and *Medicago sativa* in the semi-arid agro-pasture ecotone in North China. Based on hierarchical clustering analysis of soil volumetric water content (SWC), soil was classified into four layers (0–5 cm, 5–10 cm, 10–20 cm and 20–30 cm) as different water sources. The main sources for *Bromus inermis* were from 20–30 cm (27.0%) and groundwater (24.2%) in May, to 0–5 cm (33.9%) and 5–10 cm (26.8%) in June, became groundwater (54.7%) in July, and then to 10–20 cm and 20–30 cm in August (23.2% and 20.6%) and September (35.1% and 32.1%). *Medicago sativa* were from groundwater (52.9%) and 20–30 cm (32.4%) in May, to 0–5 cm in June (61.0%), July (39.9%), August (47.6%), and then to 5–10 cm (77.8%) in September. Regression analysis showed a negative relationship between SWC and contribution of water uptake (CWU) (CWU = −2.284 × SWC + 60.833), when the difference in water isotopes was small among soil layers. Finally, the two grassland types showed distinct soil water dynamics shaped by species-specific water use strategies and associated soil pore properties. These results indicate that water use characteristics are species specific and a species combination with less water competition is recommended for sustainable revegetation of degraded land.

**Keywords:** water use characteristics; herbaceous plants; water stable isotopes; soil volumetric water content; degraded land revegetation; agro-pasture ecotone

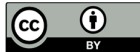

## 1. Introduction

The agro-pasture ecotone in North China (APENC) ranges more than 540,000 km² and supports a population of tens of millions [1,2]. Due to historical anthropogenic disturbance and recent climatic change, the APENC is ecologically fragile and experiences increasingly severe land degradation, water shortage, productivity reduction and biodiversity loss [3–6]. To neutralize land degradation, restore ecosystem functions and enhance biodiversity, ecological restoration measures, including revegetation (e.g., artificial grassland and plantation) and natural regeneration, have been widely implemented [7–9].

Different ecological restoration measures with diverse plant species have been adopted for rehabilitation of various types of degraded lands [10–12], and herbaceous plants exhibit outstanding performance with multiple morphological, physiological and functional traits [13,14]. Of paramount importance, herbaceous plants promote regeneration of soil fertility and productivity, with traits including quick soil cover, plentiful organic residues and biological nitrogen fixation [11,15]. Moreover, herbaceous plants also facilitate immobilization of heavy metals and other pollutants with traits of active roots

and general stress tolerance [16-18]. Additionally, herbaceous plants ensure a stable supply of forage to livestock and bring economic benefit to local inhabitants, especially for artificial grassland. However, the effects of revegetation with herbaceous plants on water conditions need further study, and it is still uncertain that grassland would improve water yield capacity or aggravate soil dry layer [10,19]. For instance, with rich genetic diversity [20], soil health maintaining ability and high nutritional feed quality [11], *Medicago sativa* (alfalfa) has been widely applied to restore degraded land; however, due to its high-water consumption, alfalfa has also caused soil desiccation and might exert potentially negative impact on regional water balance [10]. As alfalfa is an exotic plant for the APENC, to clarify its adaptability to local climates and soil types, it is necessary to compare soil water consumption and influence on soil water dynamics of alfalfa with that of natural plants (e.g., *Bromus inermis*).

Sustainable water utilization is essential for revegetation of degraded lands in regions with seasonal water shortage [12]. Different revegetation types have distinct responses to changes in soil water and hence exert varying impacts on ecosystem water balance [10,12,14]. It is therefore important to elucidate water use characteristics or water uptake patterns of plant species used for revegetation. Except for several halophytes and xerophytes [21,22], variability of stable isotope ratios of oxygen ($\delta^{18}O$) and hydrogen ($\delta^2H$) in soil water and xylem water has been widely used to estimate soil water uptake depth of plants and to identify their water use characteristics [23–25], since no isotopic discrimination occurs during water uptake by roots [26,27].

Although numerous studies using stable isotopes ($\delta^{18}O$ and/or $\delta^2H$) have reported estimation of soil water uptake depths of trees in forests [12,25,28–31], crops in croplands [32–35], shrubs in deserts [36,37], and herbaceous plants in grasslands [38–40], a large knowledge gap still exists about this issue, especially for herbaceous plants revegetating degraded lands. Recent studies have investigated the woodland expansion trend [2], wind erosion potential [1] and exotic herbaceous plant invasion [7] problems of the semi-arid agro-pasture ecotone in North China (APENC). Moreover, water use characteristics of grasses, shrubs [41,42] and trees [12] were also examined in this area. However, in these studies, plant water uptake patterns were explored in natural grasslands and artificial monoculture or mixed plantations. There is limited understanding about the water use characteristics and influences on soil water dynamics of artificial grassland species revegetating degraded land, particularly the difference between native (*Bromus inermis*) and exotic (*Medicago sativa*) species.

To remedy this deficiency, dual stable isotopes of $\delta^2H$ and $\delta^{18}O$ were used to determine water uptake sources of *Bromus inermis* and *Medicago sativa* for revegetation of degraded land in the semi-arid APENC. The objectives of this study were to: (1) identify the main water uptake layers of the two herbaceous species over the growing season, (2) quantify the contributions of various water sources to plant water uptake in different months for *Bromus inermis* and *Medicago sativa*, and (3) investigate the difference in water use characteristics and influences on soil water dynamics between *Bromus inermis* and *Medicago sativa*.

## 2. Materials and Methods

### 2.1. Site Description

This study was conducted in the Yanqing Ecological Research Station of Beijing Academy of Agricultural and Forestry Sciences (YERS-BAAFS) near the southeast border of the APENC between Beijing City and Heibei Province of China (40°27′53″ N, 115°50′23″ E) (Figure 1). The landform of YERS-BAAFS is the floodplain of Gui River, with a shallow depth of groundwater level of 0.5 m and an elevation of 501 m. It has a semi-arid continental monsoon climate with a mean air temperature of 8.4 °C and a mean annual precipitation amount of 466 mm [43]. Over 80% of the precipitation falls between May and September, overlapping with the growing season. Water shortage frequently occurs in

months between March and June due to little precipitation [1]. The soil in the study area is as thin as approximately 30 cm with the texture of sandy loam, consisting of 7% clay (<0.002 mm), 22% silt (0.002–0.02 mm) and 71% sand (0.02–2 mm). Wind erosion and dust-storm were severe in spring and early summer in this region because of loose topsoil lacking vegetation, strong wind and scarce rainfall [1]. To mitigate the environmental problems caused by regional land degradation, land use type was converted from summer maize farmland to artificial grassland in the study area in 2014. The revegetating species are native *Bromus inermis* and exotic *Medicago sativa*.

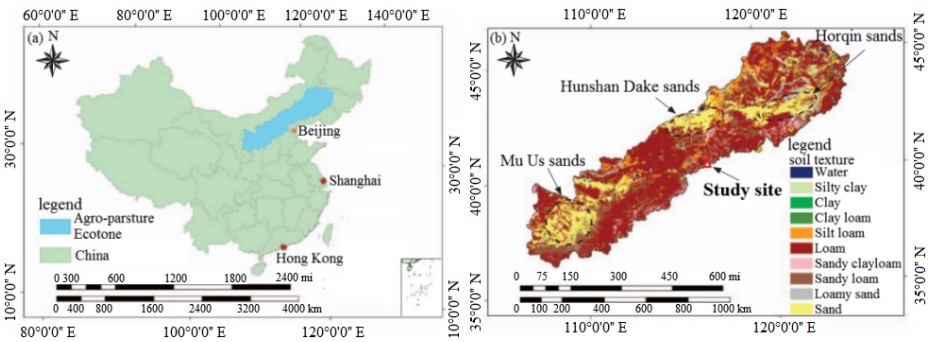

**Figure 1.** Location of study site. (**a**) The agro-pasture ecotone in North China. (**b**) The study site.

### 2.2. Sample Collection

The sampling campaigns were implemented in the grassland revegetation field at the YERS-BAAFS from May to September of 2018. In August 2014, the field was planted with *Bromus inermis* and *Medicago sativa* in contiguous plots with an equal area of 10 m × 10 m and there were 4 replicates for each type of grassland (Figure 2a). A 2 m × 2 m subplot for soil and plant sampling was set in the center of each plot and a well for groundwater sampling was set in the center of the field (Figure 2). The sampling well, with a depth of 2 m, was used for water quality monitoring and the aquifer beneath the soil horizons is pebbly sand (Figure 2b). As a kind of nature-based solutions, grasslands of *Bromus inermis* and *Medicago sativa* were left to develop naturally without irrigation and fertilization after germination of seeds except for once cutting every year in early winter. Moreover, the two types of grassland had similar plant densities (~1800 plant·m⁻²).

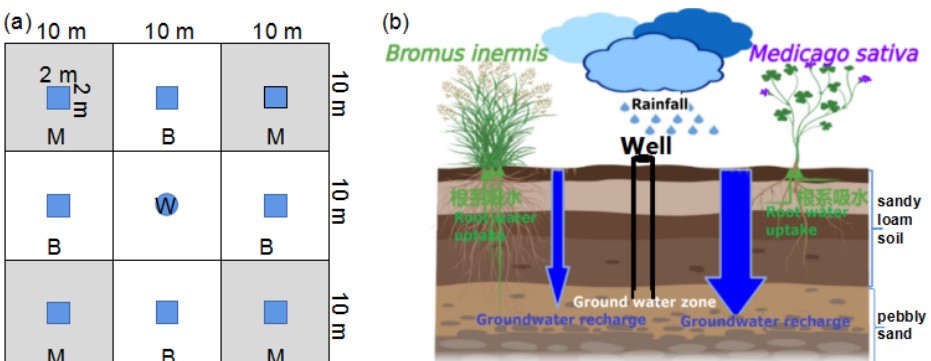

**Figure 2.** Sampling scheme. (**a**) Experimental plots (M—Medicago sativa, B—Bromus inermis, and W—Well). (**b**) Soil horizons and aquifer.

Four samples of plant xylem and soil at different depths were collected in subplots per month from May to September. There should be 20 xylem samples for each grassland type and 20 soil samples for each soil layer. As for one xylem sample, 8 to 10 individual plants of each species were randomly selected in each plot monthly, xylem was cut from selected plants and mixed together with the epidermis being carefully removed with

tweezers. The treated xylem was immediately sealed into glass vials with parafilm and stored in a freezer (−20 °C). Soil samples were collected simultaneously along the 0–30 cm soil profile with a 10 cm-diameter soil auger beside the sampled plants monthly. Soil cores were collected every 5 cm in the 0–10 cm range and every 10 cm in the 10–30 cm range. The soil from each layer was well mixed, sealed into glass vials with parafilm and kept frozen (−20 °C). Groundwater samples were obtained from a well at the study site. Additionally, a total of 18 rainwater samples were collected into 500 mL opaque air-tight glass bottles using a funnel with a ping-pong ball after rainfall.

### 2.3. Measurement and Analysis

Xylem and soil samples were extracted using an automatic cryogenic vacuum condensation system (LI-2100, LICA, Beijing, China). Over 98% of water was extracted from samples. If the sample was too dry to extract enough water for measurement, then extracted water of 2 replicating samples would be merged as one sample. So the number of xylem and soil water samples might be less than 20. The $\delta^2H$ and $\delta^{18}O$ of xylem water, soil water, groundwater and rainwater were measured using a water isotope analyzer (WIA-35d-EP, Model 912–0026, Los Gatos Research, Mountain View, CA, USA). The precision was ± 0.5‰ and ± 0.15‰ for $\delta^2H$ and $\delta^{18}O$, respectively. Each sample was measured six times, and the first two results were cast away due to memory effect. To minimize the effects of methanol and ethanol contamination, $\delta^2H$ and $\delta^{18}O$ of xylem water samples were corrected with a standard curve [44]. Additionally, SWC at depths of 5 cm, 10 cm, 15 cm, 20 cm and 25 cm was monitored at 30 min intervals using a 5TM probe connected to a Decagon EM50 data logger (METER Group, Inc., Pullman, WA, USA) in the field.

Both soil water and groundwater are primary plant water sources on the floodplain because plant roots can reach the groundwater due to its shallow buried depth. The stable isotopic composition of soil water depends on depth and becomes more enriched up to the surface layer due to the fractionation effect of evaporation. The stable isotopic composition of groundwater depends on the recharging river water and has little change during the year. The groundwater in aquifer can influence the stable isotopic composition of soil water in the bottom layer through capillary action. Xylem water is a mixture of uptake sources depending on the root characteristics of plant types

The Bayesian mixing model MixSIAR (version 3.1.7) was employed to determine the contribution of each water source to plant water uptake based on the mass balance of the isotopes [45,46].

### 2.4. Statistical Analysis

The Shapiro–Wilk test and the Levene test were used to check the distribution normality and homogeneity of variables, respectively. To classify the soil layers, SWC data of different depths were standardized with z-score normalization and clustered using Ward's method according to Euclidean distance. Significant differences in SWC, $\delta^2H$ and $\delta^{18}O$ at different soil layers and between grasslands of *Bromus inermis* and *Medicago sativa* at each soil layer were tested by one-way analysis of variance with Duncan's post hoc test ($p < 0.05$). All statistical analyses were conducted with SPSS software (SPSS Inc. Chicago, IL, USA).

## 3. Results

### 3.1. Meteorological Condition

Temporal variations in air temperature, precipitation and accumulated precipitation through the year are shown in Figure 3. The mean air temperature during the experimental period (from May to September) was 21.7 C, while the mean air temperature over the year was much lower (9.9 °C). The total precipitation from May to September was

423.2 mm, accounting for 91.4% of annual accumulated precipitation (Figure 3b). The concurrence of high air temperature and strong precipitation appeared in the experimental period, while the rest of the year had lower air temperature and little precipitation.

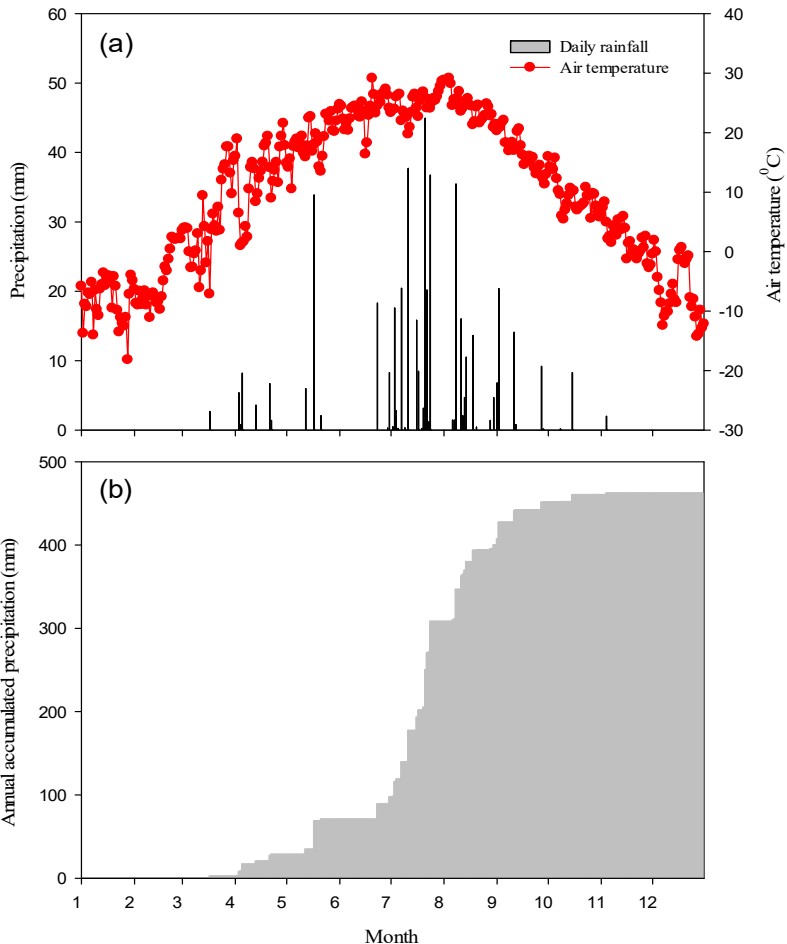

**Figure 3.** Trends of air temperature and precipitation. (**a**) Daily precipitation and air temperature. (**b**) Accumulated precipitation.

### 3.2. Soil Water Dynamics and Clustering Analysis

Soil volumetric water content (SWC) at 5 cm, 10 cm, 15 cm, 20 cm and 25 cm in grasslands of *Bromus inermis* and *Medicago sativa* had similar temporal trends of an initial decrease from May to June, then an increase from June to July, and successive decrease from July to September, with the highest SWC values in July and the lowest in June or September (Figure 4a,c; Table 1). The SWC temporal trends were in accordance with that of the precipitation amount (Figure 3; Table 1). Moreover, the stratification of SWC was evident through the experimental period for both grasslands (Figure 4a,c). The SWC firstly decreased, then increased and decreased again with soil depth in *Bromus inermis* grassland, being greatest at 20 cm ($p < 0.05$) and lowest at 10 cm. In contrast, SWC constantly increased with soil depth in *Medicago sativa* grassland, being greatest at 25 cm ($p < 0.05$) and lowest at 5 cm (Figure 4b,d; Table 1). Finally, at 5 cm, 10 cm, 15 cm and 20 cm, SWC values were significantly higher ($p < 0.05$) in *Bromus inermis* grassland than in *Medicago sativa* grassland from May to September (Figure 4b,d; Table 1). At 25 cm, SWC values were also significantly higher ($p < 0.05$) in *Bromus inermis* grassland than in *Medicago sativa* grassland in May and June, but in July and August, SWC values in

*Bromus inermis* grassland were significantly lower ($p < 0.05$) than that in *Medicago sativa* grassland, and the difference of SWC values in September and from May to September were not significant ($p = 0.067$ and $0.814$) between the two types of grassland (Table 1).

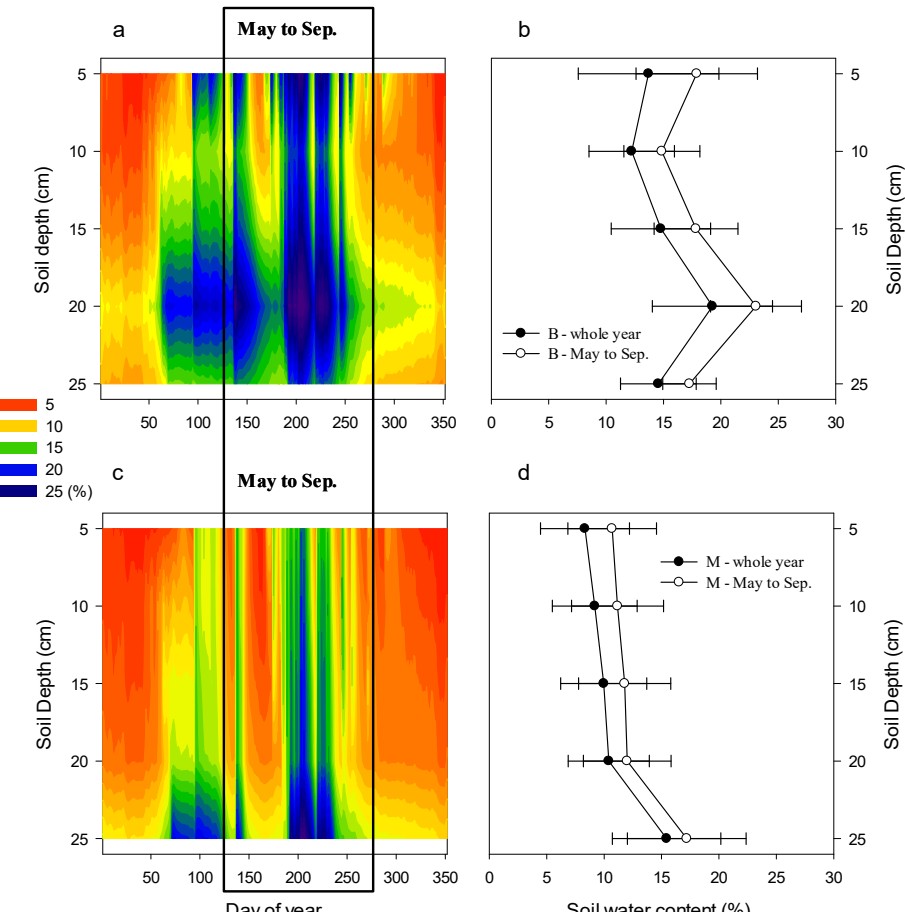

**Figure 4.** Temporal-spatial variation of soil volumetric water content. Soil volumetric water content (SWC) in the *Bromus inermis* grassland (**a**,**b**) and the *Medicago sativa* grassland (**c**,**d**). Different colors indicate different soil volumetric water content (**a**,**c**). Mean SWC at each soil depth for the whole year and for the period from May to September are depicted on the right-side (**b**,**d**). Bars denote standard deviation.

**Table 1.** Soil water dynamics at different depths from May to September in *Bromus inermis* and *Medicago sativa* grasslands.

| Grassland Types | Month | Soil Volumetric Water Content, SWC (%) | | | | | | | | | | | | | | | | | | | |
| | | 5 cm | | | | 10 cm | | | | 15 cm | | | | 20 cm | | | | 25 cm | | | |
| | | Max | Min | Avg | SD | Max | Min | Avg | SD | Max | Min | Avg | SD | Max | Min | Avg | SD | Max | Min | Avg | SD |
| *Bromus inermis* | May | 25.6 | 12.1 | 17.9 aAA | 3.7 | 19.6 | 13.2 | 15.2 bAA | 1.7 | 23.0 | 16.6 | 18.5 aAA | 1.7 | 28.5 | 21.9 | 24.2 cAA | 1.9 | 19.5 | 16.4 | 17.5 aAA | 1.0 |
| | June | 22.7 | 8.8 | 12.8 aBA | 3.6 | 15.4 | 10.7 | 12.1 aBA | 1.3 | 18.2 | 14.0 | 15.2 bBA | 1.3 | 24.2 | 19.0 | 20.8 cBA | 1.7 | 17.9 | 15.2 | 16.4 dBA | 0.8 |
| | July | 28.4 | 16.2 | 23.6 aCA | 3.2 | 22.8 | 11.6 | 18.5 bCA | 2.9 | 25.7 | 14.0 | 21.6 cCA | 3.4 | 30.0 | 18.8 | 26.3 dCA | 4.0 | 23.1 | 15.0 | 19.0 bCA | 2.9 |
| | August | 26.6 | 13.8 | 19.7 aAA | 4.7 | 20.5 | 12.2 | 16.1 bAA | 2.9 | 23.7 | 14.7 | 19.6 aAA | 3.0 | 28.8 | 20.0 | 25.4 cACA | 2.7 | 20.3 | 16.9 | 18.9 aCA | 0.9 |
| | September | 26.0 | 11.0 | 15.4 aDA | 3.8 | 18.7 | 10.2 | 12.3 bBA | 2.2 | 20.3 | 12.1 | 14.3 aBA | 2.2 | 24.6 | 15.7 | 18.6 cDA | 2.5 | 16.7 | 12.6 | 14.6 aDA | 1.4 |
| | May–September | 28.4 | 8.8 | 17.9 aA | 5.3 | 22.8 | 10.2 | 14.9 bA | 3.3 | 25.7 | 12.1 | 17.9 aA | 3.6 | 30.0 | 15.7 | 23.1 cA | 3.9 | 23.1 | 12.6 | 17.3 aA | 2.3 |
| *Medicago sativa* | May | 16.0 | 5.8 | 9.4 aAB | 3.0 | 16.2 | 6.8 | 9.7 abAB | 2.8 | 16.8 | 7.6 | 10.3 abAB | 2.6 | 16.8 | 8.7 | 10.9 bAB | 2.3 | 22.0 | 13.1 | 16.1 cAB | 2.7 |
| | June | 14.7 | 4.5 | 7.7 aBB | 3.1 | 13.9 | 5.0 | 7.2 aBB | 2.5 | 11.9 | 6.3 | 7.5 aBB | 1.5 | 8.7 | 7.7 | 8.1 aBB | 0.3 | 13.0 | 11.9 | 12.3 bBB | 0.3 |
| | July | 18.9 | 11.1 | 15.3 aCB | 1.9 | 20.1 | 9.7 | 15.8 aCB | 2.4 | 21.2 | 8.1 | 16.4 aCB | 3.1 | 21.2 | 8.5 | 16.3 aCB | 3.6 | 29.1 | 12.7 | 22.2 bCB | 6.1 |
| | August | 16.5 | 7.9 | 12.1 aDB | 3.1 | 17.8 | 9.2 | 13.3 aDB | 3.0 | 18.5 | 10.1 | 14.4 bDB | 2.8 | 18.5 | 10.6 | 14.8 bDB | 2.6 | 26.0 | 15.4 | 21.3 cCB | 3.5 |
| | September | 15.4 | 5.8 | 8.9 aABB | 2.5 | 16.3 | 6.9 | 9.8 abAB | 2.5 | 14.9 | 7.8 | 10.4 bAB | 2.0 | 12.2 | 8.4 | 9.8 abAB | 1.1 | 15.2 | 12.8 | 14.0 cBA | 0.8 |
| | May–September | 18.9 | 4.5 | 10.7 aB | 3.9 | 20.1 | 5.0 | 11.2 abB | 4.0 | 21.2 | 6.3 | 11.8 bB | 4.0 | 21.2 | 7.7 | 12.0 bB | 3.8 | 29.1 | 11.9 | 17.2 cA | 5.2 |

Notes: Max, Min, Avg, and SD are abbreviations of the maximum, minimum, average and standard deviation, respectively. Superscripts of average values are results of one-way analysis of variance with Duncan's post hoc test with a significance level of 0.05, and different lowercase, uppercase and italic bold-type uppercase letters indicate significant difference in SWC between depths for the same month and grassland type, between months for the same depth and grassland type, and between grassland types for the same depth and month, respectively.

According to the hierarchical clustering analysis of SWC, soil layers were further classified specifically (Figure 5). The soil depths of both grasslands were classified into three groups (Group I, Group II and Group III); however, the group constitution was different between grasslands of *Bromus inermis* and *Medicago sativa*. In *Bromus inermis* grassland, Group I contained 5 cm and 10 cm, 15 cm and 20 cm were classified as Group II, and only 25 cm belonged to Group III. While in *Medicago sativa* grassland, Group I included only 5 cm, Group II contained 10 cm and 15 cm, and 20 cm and 25 cm fell into Group III. In practice, 15 cm was more closely connected with 20 cm than 10 cm in *Bromus inermis* grassland, but it was just the reverse for spatial connection in *Medicago sativa* grassland. As a result, the soil layers were rearranged into four parts as follows: (1) 0–5 cm; (2) 5–10 cm; (3) 10–20 cm and (4) 20–30 cm.

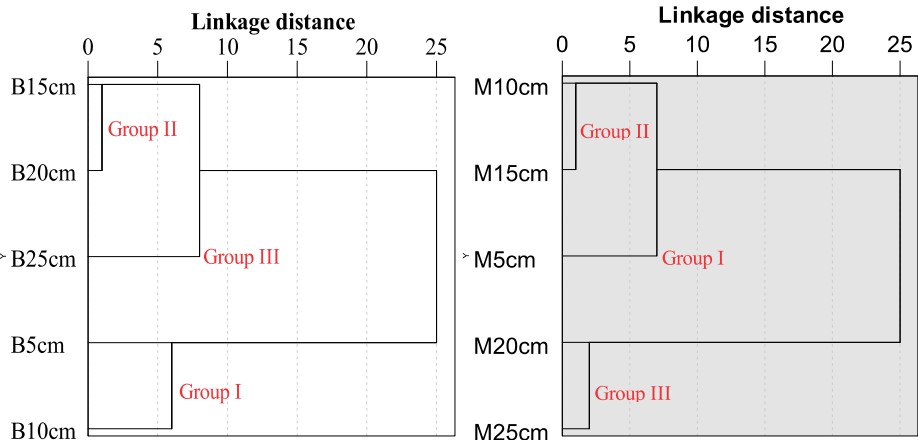

**Figure 5.** Results of hierarchical cluster analysis of soil volumetric water content. The white area (B) indicates the *Bromus inermis* grassland and the gray area (M) indicates the *Medicago sativa* grassland.

### 3.3. Isotopic Composition of Precipitation, Groundwater, Xylem Water and Soil Water

The isotopic composition of precipitation showed a broad range from −14.00 to −1.86‰ for δ$^{18}$O and −100.76 to −15.49‰ for δ$^2$H, with average values of −7.52‰ and −49.98‰, respectively (Table 2). Compared with the global meteoric water line (GMWL) δ$^2$H = 8 × δ$^{18}$O + 10 [47], the local meteoric water line (LMWL) δ$^2$H = 6.73 × δ$^{18}$O + 0.62 ($R^2$ = 0.88, $p < 0.01$) obtained from local precipitation samples had a lower slope (6.73 V.S. 8) (Figure A1, Appendix A). However, for groundwater, the δ$^{18}$O and δ$^2$H varied within a narrower range from −11.48 to −9.96‰ and from −73.51 to −70.99‰, with average values of −10.51‰ and −72.06‰, respectively (Table 2; Figure 6).

**Table 2.** General characteristics of the isotopic composition in precipitation, groundwater, soil water and xylem water samples.

| Plant Species | Sample Type | Soil Depth (cm) | N | δ$^{18}$O(‰) | | | | δ$^2$H(‰) | | | |
|---|---|---|---|---|---|---|---|---|---|---|---|
| | | | | Max | Min | Avg | SD | Max | Min | Avg | SD |
| *Bromus inermis* | Soil water | 0–5 | 16 | 9.39 | −10.76 | −2.98 | 6.73 | −8.06 | −74.07 | −44.96 | 20.86 |
| | | 5–10 | 19 | 1.95 | −8.49 | −4.14 | 3.77 | −15.72 | −60.92 | −44.00 | 16.53 |
| | | 10–20 | 18 | −1.18 | −9.73 | −5.47 | 2.65 | −29.41 | −68.80 | −50.65 | 13.56 |
| | | 20–30 | 19 | −5.49 | −10.16 | −8.09 | 1.55 | −44.20 | −74.58 | −63.37 | 8.49 |
| | | Total | 72 | 9.39 | −10.76 | −5.26 | 4.37 | −8.06 | −74.58 | −50.99 | 16.91 |
| | Xylem water | | 15 | −0.55 | −8.60 | −4.86 | 3.03 | −39.55 | −69.99 | −55.60 | 12.42 |
| *Medicago sativa* | Soil water | 0–5 | 14 | 9.42 | −10.59 | −4.29 | 5.67 | −9.96 | −74.48 | −47.20 | 18.39 |
| | | 5–10 | 17 | 5.62 | −8.86 | −4.01 | 4.87 | −11.60 | −66.39 | −44.64 | 18.68 |
| | | 10–20 | 19 | −2.09 | −9.69 | −6.25 | 2.87 | −30.61 | −68.96 | −52.52 | 13.32 |

| | | Max | Min | Avg | SD | | Max | Min | Avg | SD |
|---|---|---|---|---|---|---|---|---|---|---|
| | 20–30 | 19 | −3.68 | −10.54 | −7.90 | 1.64 | −44.59 | −72.64 | −64.08 | 6.86 |
| | Total | 69 | 9.42 | −10.59 | −5.75 | 4.14 | −9.96 | −74.48 | −52.68 | 16.27 |
| Xylem water | | 14 | 2.41 | −8.52 | −4.52 | 3.34 | −38.29 | −65.50 | −55.79 | 9.68 |
| Groundwater | | 16 | −9.96 | −11.48 | −10.51 | 0.38 | −70.99 | −73.51 | −72.06 | 0.76 |
| Precipitation | | 18 | −1.86 | −14.00 | −7.52 | 2.95 | −15.49 | −100.76 | −49.98 | 21.11 |

Note: Max, Min, Avg, and SD are abbreviations of the maximum, minimum, average and standard deviation, respectively.

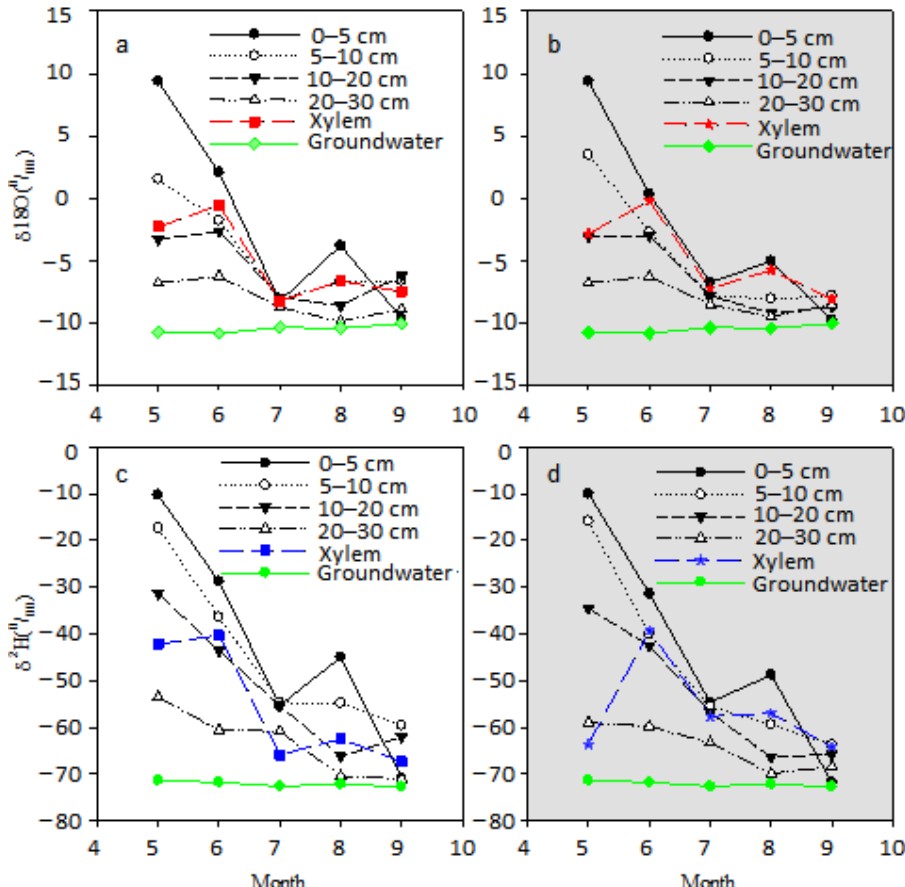

**Figure 6.** Monthly trends of $\delta^2H$ and $\delta^{18}O$ of soil water, xylem water and groundwater. Variation in $\delta^{18}O$ (**a,b**) and $\delta^2H$ (**c,d**) in soil water, xylem water and groundwater from May to September. The white area indicates the *Bromus inermis* grassland and the gray area indicates the *Medicago sativa* grassland.

The $\delta^{18}O$ and $\delta^2H$ values of xylem water were −4.86 ± 3.03‰ and −55.60 ± 12.42‰, respectively, for *Bromus inermis*, and −4.52 ± 3.34‰ and −55.79 ± 9.68‰, respectively, for *Medicago sativa* (Table 2). The $\delta^2H$-$\delta^{18}O$ relationship in xylem water or xylem water line (XWL) could be expressed as $\delta^2H = 3.09 \times \delta^{18}O - 41.20$, $R^2 = 0.77$, $p < 0.01$. The XML for *Bromus inermis* and *Medicago sativa* were $\delta^2H = 4.01 \times \delta^{18}O - 36.15$, $R^2 = 0.95$, $p < 0.01$ and $\delta^2H = 2.29 \times \delta^{18}O - 45.43$, $R^2 = 0.62$, $p < 0.01$, respectively (Figure A1). There was no significant difference ($p = 0.962$ for $\delta^{18}O$, $p = 0.954$ for $\delta^2H$) in the xylem water isotopic composition between *Bromus inermis* and *Medicago sativa*. However, the $\delta^{18}O$ and $\delta^2H$ values of xylem water of *Bromus inermis* and *Medicago sativa* exhibited statistically significant variation with month ($p < 0.001$) (Figure 6).

There was no significant difference ($p = 0.489$ for $\delta^{18}O$, $p = 0.546$ for $\delta^2H$) in isotopic ratios of soil water between grasslands of *Bromus inermis* and *Medicago sativa*. For *Bromus inermis* grassland, the average $\delta^{18}O$ and $\delta^2H$ values of soil water were −5.26 ± 4.37‰ and

−50.99 ± 16.91‰, respectively, with the highest values of 9.39‰ (0–5 cm) and −8.06‰ (0–5 cm), respectively, and lowest values of −10.76‰ (0–5 cm) and −74.58‰ (20–30 cm), respectively (Table 2). For *Medicago sativa* grassland, the average $\delta^{18}$O and $\delta^2$H values of soil water were −5.75 ± 4.14‰ and −52.68 ± 16.27‰, respectively, and the respective highest and lowest isotopic compositions all occurred at 0–5 cm (Table 2).

According to the results of the clustering analysis of SWC, the variations in soil water isotopes were divided into four groups (Figure 6). With the increasing depth of soil layers, the isotopic values of soil water exhibited more depleted, less variable and more approximate to groundwater (Figure 6). The $\delta^{18}$O and $\delta^2$H values of soil water at 20–30 cm layer were both significantly lower than that at the other three soil layers ($p < 0.01$); however, the more enriched $\delta^{18}$O and $\delta^2$H values of soil water at 0–5 cm, 5–10 cm and 10–20 cm layers showed no significant difference ($p > 0.05$). Moreover, the $\delta^{18}$O and $\delta^2$H values of soil water did not differ significantly ($p > 0.05$) between grasslands of *Bromus inermis* and *Medicago sativa* at every soil layer.

The soil water line (SWL) derived from soil water samples could be expressed as $\delta^2$H = 3.66 × $\delta^{18}$O—31.71, $R^2$ = 0.88, $p < 0.01$. The SWL for grasslands of *Bromus inermis* and *Medicago sativa* were $\delta^2$H = 3.57 × $\delta^{18}$O—32.22, $R^2$ = 0.85, $p < 0.01$ and $\delta^2$H = 3.76 × $\delta^{18}$O—31.06, $R^2$ = 0.92, $p < 0.01$, respectively (Figure A1). SWLs varied with soil layer (Figure A2). In *Bromus inermis* grassland, the SWLs at 0–5 cm, 5–10 cm, 10–20 cm and 20–30 cm were $\delta^2$H = 3.02 × $\delta^{18}$O—35.94, $R^2$ = 0.95, $p < 0.01$, $\delta^2$H = 4.23 × $\delta^{18}$O—26.49, $R^2$ = 0.93, $p < 0.01$, $\delta^2$H = 4.15 × $\delta^{18}$O—27.95, $R^2$ = 0.66, $p < 0.01$, and $\delta^2$H = 4.21 × $\delta^{18}$O—29.35, $R^2$ = 0.59, $p < 0.01$, respectively (Figure A2). In *Medicago sativa* grassland, the SWLs at 0–5 cm, 5–10 cm, 10–20 cm and 20–30 cm were $\delta^2$H = 3.11 × $\delta^{18}$O—33.87, $R^2$ = 0.92, $p < 0.01$, $\delta^2$H = 3.78 × $\delta^{18}$O—29.49, $R^2$ = 0.97, $p < 0.01$, $\delta^2$H = 4.45 × $\delta^{18}$O—24.70, $R^2$ = 0.92, $p < 0.01$, and $\delta^2$H = 3.75 × $\delta^{18}$O—34.44, $R^2$ = 0.80, $p < 0.01$, respectively (Figure A2).

*3.4. Intersections of Isotopic Composition between Groundwater, Soil Water and Xylem Water*

Xylem water is taken to be the mixture of different water sources, according to the assumption that isotope fractionation did not occur during the uptake of water by root and the transport of water by vascular system for most plants [48,49]. Hence, the main water uptake source for plant growth can be determined by the intersection of the isotopic composition between xylem water and source water (soil water and groundwater) (Figure A3).

For *Bromus inermis* grassland, the $\delta^{18}$O of xylem water intersected with the soil layer of 10–20 cm in May, 5–10 cm in June, 10–20 cm in July, 5–10 cm in August, and 5–10 cm and 20 cm in September (Figure A3). The $\delta^2$H of xylem water intersected with the soil layer of 20 cm in May, 10–20 cm in June, deeper than 30 cm (groundwater) in July, 10–20 cm in August, and 5 cm and 20–30 cm in September (Figure A3). In September, more than one intersections were found.

For *Medicago sativa* grassland, the $\delta^{18}$O of xylem water intersected with the soil layer of 10–20 cm in May, 0–5 cm in June, 5–10 cm in July, 0–5 cm in August, and 5–10 cm in September (Figure A3). The $\delta^2$H of xylem water intersected with the soil layer of deeper than 30 cm (groundwater) in May, 5–10 cm in June,10–20 cm in July, 5–10 cm in August, and 5–10 cm in September (Figure A3).

### 3.5. Contribution of Each Water Source to Plant Water Uptake

The water uptake proportions from each water source changed over the period from May to September for *Bromus inermis* and *Medicago sativa* (Figure 7). For *Bromus inermis*, the proportional water contributions of 20–30 cm and groundwater in May were 27.0 ± 18.5% and 24.2 ± 12.7%, respectively. The main water uptake layer changed to 0–5 cm and 5–10 cm in June, with proportions of 33.9 ± 16.1% and 26.8 ± 19.5%, respectively. In July, *Bromus inermis* obtained the largest proportion of water from groundwater (54.7 ± 6.6%) again. From August to September, *Bromus inermis* mainly absorbed water at soil layers of 10–20 cm and 20–30 cm, and the proportions showed an increasing trend, from 23.2 ± 17.7% and 20.6 ± 15.1% to 35.1 ± 9.5% and 32.1 ± 15.1%, respectively. *Medicago sativa* had relatively simple variations in water uptake among water sources. *Medicago sativa* derived 52.9 ± 16.9% of its water from groundwater and 32.4 ± 21.5% from soil layer of 20–30 cm in May. The major water source was shifted to soil layer of 0–5 cm in June (61.0 ± 8.7%), July (39.9 ± 14.4%) and August (47.6 ± 10.4%). In September, the primary water uptake source changed to soil layer of 5–10 cm, with the proportion of 77.8 ± 8.8%.

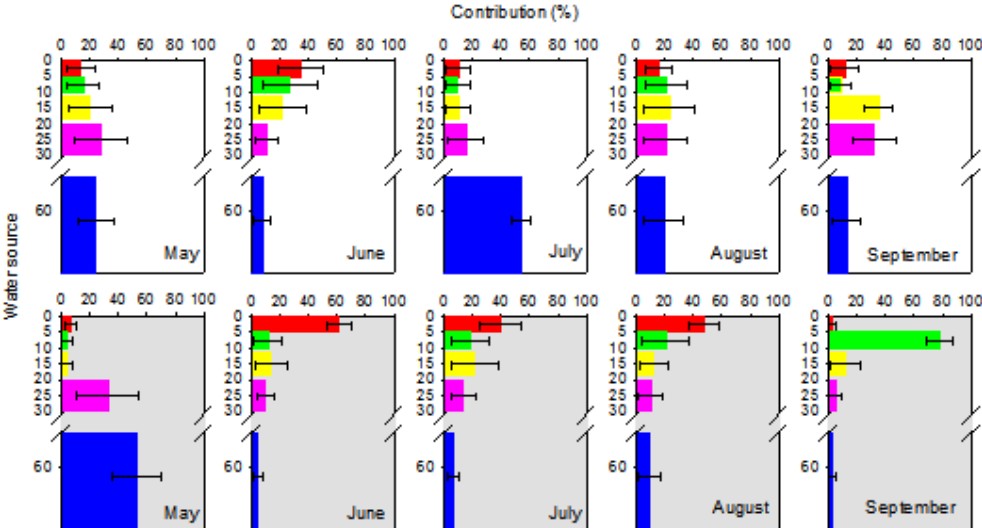

**Figure 7.** Contribution of soil water and groundwater to plant water uptake. Water uptake of *Bromus inermis* (white area) and *Medicago sativa* (gray area) from 0–5 cm, 5–10 cm, 10–20 cm, and 20–30 cm soil layers (above breaks) and ground water (below breaks) in different months. Bars indicate standard deviation.

## 4. Discussion

### 4.1. Water Source Partitioning in the Artificial Grasslands

Partitioning the water source is necessary before determining plant water uptake patterns. The isotopic composition of groundwater was significantly lower than that of soil water ($p < 0.01$) (Figure 6; Table 2). In addition, the soil water isotopes at 20–30 cm layer were more depleted than that at other shallower soil layers ($p < 0.05$), which were not significantly different among themselves ($p > 0.05$) (Figure 6; Table 2), suggesting that the 20–30 cm soil layer differentiated from all other soil layers. Considering its assets of accessibility, high accuracy and adjustable monitoring frequency, the SWC was selected as an indicator to further partition soil layers with clustering analysis [35]. Combining the results of water isotopic composition and clustering analysis of SWC, water sources for plant uptake were divided into five parts: groundwater and soil layers of 0–5 cm, 5–10 cm, 10–20 cm and 20–30 cm.

The partitioning of water source for plant water uptake depends on soil and plant properties. Soil attributes, including soil texture and bulk density, determined plant water

uptake by adjusting water motion and holding capacity along soil profiles [50]. Soil texture and groundwater depth are similar for the adjoining grasslands of *Bromus inermis* and *Medicago sativa*. Thus, it is reasonable to utilize the same water source partitioning scheme for the two grassland types. Moreover, plant properties, such as root type, root morphology, density, height and stem diameter, also regulated water source for plant water uptake. In this study, plant characteristics of density, height and stem diameter are consistent for *Bromus inermis* and *Medicago sativa*. Nevertheless, root traits of *Bromus inermis* are different from that of *Medicago sativa*. Root length and root surface area of *Bromus inermis* with fibrous root system are higher than that of *Medicago sativa* with taproot system, while root volume of *Bromus inermis* is lower than that of *Medicago sativa* [51,52].

### 4.2. Plant Water Use Characteristics in the Different Types of Grassland

Plant water uptake patterns were distinct between *Bromus inermis* and *Medicago sativa*. Contributions of different water sources to water uptake for *Bromus inermis* and *Medicago sativa* changed over the months from May to September (Figure 7). These are associated with disparate root distributions between *Bromus inermis* and *Medicago sativa*, and with variations in supplying abilities of water sources and plant water requirements in different months. Although different in proportions of water uptake originated from water sources, *Bromus inermis* and *Medicago sativa* both exhibited an analogous trend in which they firstly mainly absorbed groundwater and deep soil water (20–30 cm) in May and then shallow soil water (0–5 cm and 5–10 cm) in June. However, from July to September, *Medicago sativa* predominantly depended on shallow soil water, while *Bromus inermis* mainly relied on groundwater or middle (10–20 cm) and deep soil water. This result was inconsistent with *Achnatherum splendens* and *Leymus chinensis* in grassland [40] as well as *Zea mays* L. in cropland [35], where plants depended on shallow soil water initially, then shifted their water source to deep soil water, and finally returned to shallow soil water. This discrepancy may be caused by the condition that deep soil water and groundwater were more available than shallow soil water due to a long-time rainless period with intense evaporation in May, and that the perennial plants *Bromus inermis* and *Medicago sativa* had deeply distributed roots for the consumption of deep soil water and groundwater in the early dry season.

The dichotomization of water uptake patterns from July to September between *Bromus inermis* and *Medicago sativa* may be explained by different root allocation modes. *Bromus inermis* was prone to exploiting deep soil water and groundwater, which was largely a consequence of its deeper root distribution with greater root mass, root length, root tips and root forks in the subsoil layer than in the topsoil layer [52–54]. At the same time, *Medicago sativa* was reversely apt to consume shallow soil water in the rainy wet season, which probably resulted from its larger proportion of root surface area, root tips, root length and root volume of lateral and fine roots (with diameters no more than 2 mm) being located in the topsoil layer than in the subsoil layer [52,55]. These results indicated that *Medicago sativa* showed a more opportunistic water use pattern depending heavily on capturing shallow soil water from sporadic precipitation moisture. In contrast, *Bromus inermis* exhibited a more adaptive water use strategy taking advantage of deep soil water and groundwater. This could allow *Bromus inermis* to avert water competition with *Medicago sativa* absorbing shallow soil water [54,56].

Determining plant water uptake source with isotopic composition comparison would be less powerful when the difference in soil water isotopes was small along the soil profile for some periods (such as July in this study, Figure 6). As a result, soil water content was considered to determine plant water uptake source [25,35]. For grasslands of *Bromus inermis* and *Medicago sativa*, the negative correlation between the contribution of water uptake (CWU) and the soil volumetric water content (SWC) in July, CWU = −2.284 × SWC + 60.833, r = 0.716, $p$ = 0.046, $n$ = 8, implied that soil could be desiccated by plant water uptake.

### 4.3. Influences of Land Restoration on Soil Water Dynamics

Ecological restoration measures exert influences on soil water dynamics directly by plant attributes determining water uptake and indirectly by soil properties relating to water holding and infiltration capacities. In this study, the similar phenology and trends of transpiration between *Bromus inermis* and *Medicago sativa* resulted in identical temporal trends of SWC at various depths for grasslands of *Bromus inermis* and *Medicago sativa* (Table 1), while the distinct rooting depths between *Bromus inermis* and *Medicago sativa* led to different spatial distribution patterns of SWC along the soil profiles (Figure 4; Table 1). This result implies that plant species for ecological restoration regulate temporal-spatial variation in SWC via species-specific strategies of water acquisition [57]. In addition, the differentiation of soil features between grasslands of *Bromus inermis* and *Medicago sativa* also plays a critical role in adjusting soil water dynamics. Specifically, the soil organic carbon content (SOC) and soil capillary porosity were higher in *Medicago sativa* grassland than in *Bromus inermis* grassland after 4 years of revegetation, and the difference was statistically significant for SOC at shallow soil layers (0–5 cm and 5–10 cm) (Table 3). These may contribute to the noticeably higher 25 cm-depth SWC in *Medicago sativa* grassland than in *Bromus inermis* grassland in July and August (Table 1). The results could be explained by the eco-hydrological separation hypothesis, meaning that tightly bound water and mobile water, as separated soil water components, supplies plant transpiration or soil evaporation and recharges groundwater, respectively [58–60]. Here, SOC, porosity and infiltration capacity closely and positively related to each other [61,62]. Surface soil infiltration rate (SIR) was decreased in *Bromus inermis* grassland by its lower SOC and fibrous, rhizomatous roots tending to compact the soil and block the water flow paths [63]. However, SIR was increased in *Medicago sativa* grassland due to the higher SOC and the interpenetrating and connecting effects of the tap roots on soil pores [51,64,65]. Hence, for the shallow soil layers, large pores, through which a greater proportion of rainwater as mobile water passed to reach the deep soil layer, were dominant in *Medicago sativa* grassland. In contrast, small pores, retaining more rainwater as tightly bound water, prevailed in *Bromus inermis* grassland and little rainwater percolated through the profile to the deep soil layer. So depending on rainfall amount, the heavy rainfall events moistened the deep soil layer more in *Medicago sativa* grassland than in *Bromus inermis* grassland during the rainy July and August [35,37].

**Table 3.** Soil organic carbon content and soil capillary porosity in Bromus inermis and Medicago sativa grasslands.

| Grassland Type | Soil Depth (cm) | N | Soil Organic Carbon Content (g·kg⁻¹) | | | | Soil Capillary Porosity (%) | | | |
|---|---|---|---|---|---|---|---|---|---|---|
| | | | Max | Min | Avg | SD | Max | Min | Avg | SD |
| *Bromus inermis* | 0–5 | 4 | 22.11 | 16.67 | 20.18 [aA] | 2.40 | 36.52 | 34.16 | 35.37 [A] | 1.02 |
| | 5–10 | 4 | 17.94 | 15.01 | 16.26 [bA] | 1.32 | | | | |
| | 10–20 | 4 | 10.67 | 7.43 | 8.84 [cA] | 1.53 | | | | |
| | 20–30 | 4 | 8.53 | 3.02 | 5.28 [dA] | 2.61 | | | | |
| | Total | 4 | 22.11 | 3.02 | 12.64 [A] | 6.35 | | | | |
| *Medicago sativa* | 0–5 | 4 | 29.45 | 21.38 | 26.19 [aB] | 3.79 | 41.09 | 34.49 | 37.19 [A] | 2.98 |
| | 5–10 | 4 | 24.58 | 17.30 | 21.83 [aB] | 3.15 | | | | |
| | 10–20 | 4 | 12.38 | 7.47 | 10.07 [bA] | 2.01 | | | | |
| | 20–30 | 4 | 15.51 | 4.17 | 9.96 [bA] | 6.02 | | | | |
| | Total | 4 | 29.45 | 4.17 | 17.01 [B] | 8.23 | | | | |

Notes: Max, Min, Avg, and SD are abbreviations of the maximum, minimum, average and standard deviation, respectively. Superscripts of average values are results of one-way analysis of variance with Duncan's post hoc test with a significance level of 0.05, and different lowercase and uppercase letters indicate significant difference in SWC between depths in the same grassland type and between grassland types at the same depth, respectively.

Different patterns of soil water distribution and water compartmentalization between grasslands of *Bromus inermis* and *Medicago sativa* may further influence processes of runoff generation, groundwater recharging and evaporation-transpiration partitioning [60]. First of all, superfluous rainwater tended to form surface runoff in *Bromus inermis* grassland but inclined to replenish ground water in *Medicago sativa* grassland, due to different soil infiltration capacities. Additionally, the ratio of evaporation to transpiration would be higher in *Bromus inermis* grassland than in *Medicago sativa* grassland, because the significantly moister topsoil in *Bromus inermis* grassland (Figure 4; Table 1) promoted soil evaporation, which decreases rapidly with depth [66]. In accordance with previous studies, our results demonstrated that deep-rooted *Bromus inermis*, as a native plant, can occupy available rooting space rapidly but inhibit water infiltration into the soil at the same time [51–53], while shallow-rooted *Medicago sativa*, as an exotic plant, can fertilize and desiccate topsoil effectively but facilitate water infiltration to recharge groundwater as well [11,51,52]. Consequently, to realize sustainable water resources utilization and ecosystem water balance, mixed planting *Bromus inermis* and *Medicago sativa* would be more desirable for revegetation of degraded lands in areas with both strong precipitation events and seasonal water shortage.

### 4.4. Implications

As a climatic and ecological transition zone, the semi-arid agro-pasture ecotone in North China (APENC) has experienced a rise in temperature and decrease in precipitation over the past 50 years [67] and may be highly sensitive to climate oscillations in the future [68]. Meanwhile, the APENC has also experienced a large-scale ecological restoration, especially grassland establishment or rehabilitation, due to programs of Beijing-Tianjin Sandstorm Source Control, Northern China's Vegetation Belt, and "Grain for Green" since 2000 [69–71]. Together with the warmer and drier climate, grassland expansion will result in soil desiccation and water shortage aggravation [72]. The results here showed that *Bromus inermis* and *Medicago sativa* for degraded land revegetation had significantly different water use characteristics and influences on soil water dynamics. One aspect was that surface soil desiccation was more serious in *Medicago sativa* grassland than in *Bromus inermis* grassland, with 0–20 cm SWC of 11.4% and 18.4% (May to September), respectively (Figure 4; Table 1). Another aspect was water source segregation between the two plant species, with deeper water sources of *Bromus inermis* than *Medicago sativa* (Figure 7). Finally, temporal-spatial variation in SWC was regulated by plant species selected for revegetation based on eco-hydrological separation, and soil evaporation and groundwater recharging were facilitated by grasslands of *Bromus inermis* and *Medicago sativa*, respectively. Our results indicated that interspecific competition for water source might not occur between *Bromus inermis* and *Medicago sativa*, and hence they could be accreted for revegetation in water-limited areas. In addition, an anomalous climate with more extreme precipitation [73] and warmer temperature will challenge the persistence of and undermine the sustainability of revegetation in degraded lands [74]. The combination of *Bromus inermis* and *Medicago sativa* may provide a sustainable ecological restoration strategy by optimizing plant water consumption.

In this study, we only investigated the water use characteristics and influences on soil water dynamics of two typically widely used herbaceous plant species in artificial grasslands. The water use patterns of plant species of other revegetating types in the APENC, such as mixed grass–tree and mixed grass–shrub, should be explored in the future. Moreover, plant density, plant water-use efficiency and field management practice (mulching for example) are also significant for sustainable revegetation of degraded lands in semi-arid areas, because these factors help to attune balance between land restoration and water resource conservation [12,74]. Additionally, the sampled groundwater is recharged by the nearby Gui River and its isotopic composition (Table 2) falls within the range of groundwater in Beijing area (−92‰ to −52‰ for $\delta^2H$ and −13.2‰ to −6‰ for $\delta^{18}O$)

[75]. The mixed isotopic composition between groundwater and 0–5 cm soil water in September may be caused by a precipitation event, in that the rainwater, with a stable isotopic composition (−74.87‰ for $\delta^2$H and −11.98‰ $\delta^{18}$O, Figure A1) similar to groundwater (Table 2), replenished 0–5 cm soil layer. However, the contribution of 0–5 cm soil water and groundwater to *Bromus inermis* and *Medicago sativa* water consumption in September was small (Figure 7), so the mixed isotopic composition did not exert significant influence on water uptake source analysis.

## 5. Conclusions

In this study, stable isotopes of $\delta^2$H and $\delta^{18}$O were applied to explore the seasonal water use characteristics and influences on soil water dynamics of *Bromus inermis* and *Medicago sativa*, widely used for revegetation of degraded lands in the semi-arid agro-pasture ecotone in North China. As an important water source, soil was divided into four groups, 0–5 cm, 5–10 cm, 10–20 cm, and 20–30 cm, according to the hierarchical clustering analysis of SWC. The results showed that *Bromus inermis* mainly absorbed deep-layer (below 10 cm) soil water and groundwater, while *Medicago sativa* depended largely on shallow layer (0–5 cm and 5–10 cm) soil water, implying that these species had water source segregation. Moreover, variations in main water uptake sources occurred in different months: May (20–30 cm and groundwater), June (0–5 cm and 5–10 cm), July (groundwater), August and September (10–20 cm and 20–30 cm) for *Bromus inermis*, and May (groundwater and 20–30 cm), June, July and August (0–5 cm), September (5–10 cm) for *Medicago sativa*. The soil volumetric water content was negatively correlated with the water uptake contribution when isotopic composition comparison was less powerful for determining plant water uptake source. Finally, influenced by plant water use strategy and soil pore property, distinct soil water dynamics were exhibited between grasslands of *Bromus inermis* and *Medicago sativa*. These findings suggest that species-specific water use characteristics existed, and the combination of *Bromus inermis* and *Medicago sativa* was proposed for sustainable revegetation in the APENC. This study provides more accurate and widely adapted information and insights regarding the revegetation of degraded land and the ecological management related to sustainable water use in the semi-arid areas facing climatic change and anthropogenic disturbance.

**Author Contributions:** Conceptualization, Z.P. and H.K.; methodology, Z.P.; software, H.X.; validation, C.C., G.Z. and X.F.; formal analysis, Z.P.; investigation, G.Z.; resources, J.W.; data curation, Z.P.; writing—original draft preparation, Z.P.; writing—review and editing, H.K.; visualization, H.X.; supervision, X.F.; project administration, H.K.; funding acquisition, J.W. All authors have read and agreed to the published version of the manuscript.

**Funding:** This research was funded by Beijing Academy of Agriculture and Forestry Sciences, grant number QNJJ202003, KJCX20200301, by Ministry of Agriculture and Rural Affairs of the People's Republic of China, World Bank and Global Environment Facility, grant number P166853/CSMG-C-05 and by Beijing Science Foundation, grant number 5204031.

**Institutional Review Board Statement:** Not applicable.

**Informed Consent Statement:** Not applicable.

**Data Availability Statement:** The datasets generated during and/or analyzed during the current study are available from the corresponding author on reasonable request.

**Acknowledgments:** This work was supported by the Beijing Academy of Agriculture and Forestry Sciences [No. QNJJ202003 and KJCX20200301], Climate Smart Management of Grassland Ecosystems (Project ID: P166853/CSMG-C-05) from Ministry of Agriculture and Rural Affairs of the People's Republic of China, World Bank and Global Environment Facility and Beijing Science Foundation [No. 5204031].

**Conflicts of Interest:** The authors declare no conflict of interest.

## Appendix A

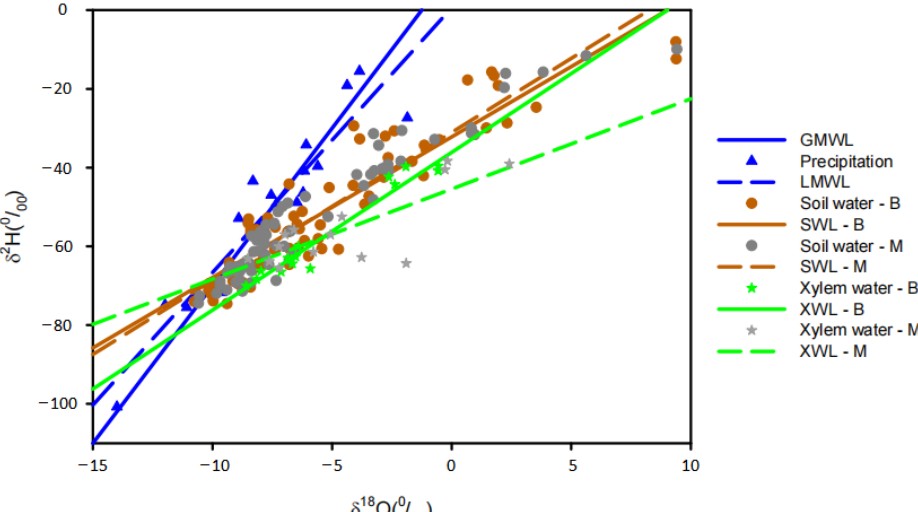

**Figure A1.** δ²H-δ¹⁸O relationship in precipitation, groundwater, soil water, and xylem water. The solid blue triangle denotes precipitation. The brown and gray solid circles denote the soil water of *Bromus inermis* (B) and *Medicago sativa* (M) grasslands, respectively. The green and gray stars denote the xylem water of *Bromus inermis* (B) and *Medicago sativa* (M), respectively. The blue solid line, blue dash line, brown solid line, brown dash line, green solid line and green dash line represent the global meteoric water line (GMWL), local meteoric water line (LMWL), soil water line (SWL) of *Bromus inermis* grassland (B), SWL of *Medicago sativa* grassland (M), xylem water line (XWL) of *Bromus inermis* (B) and XWL of *Medicago sativa* (M), respectively.

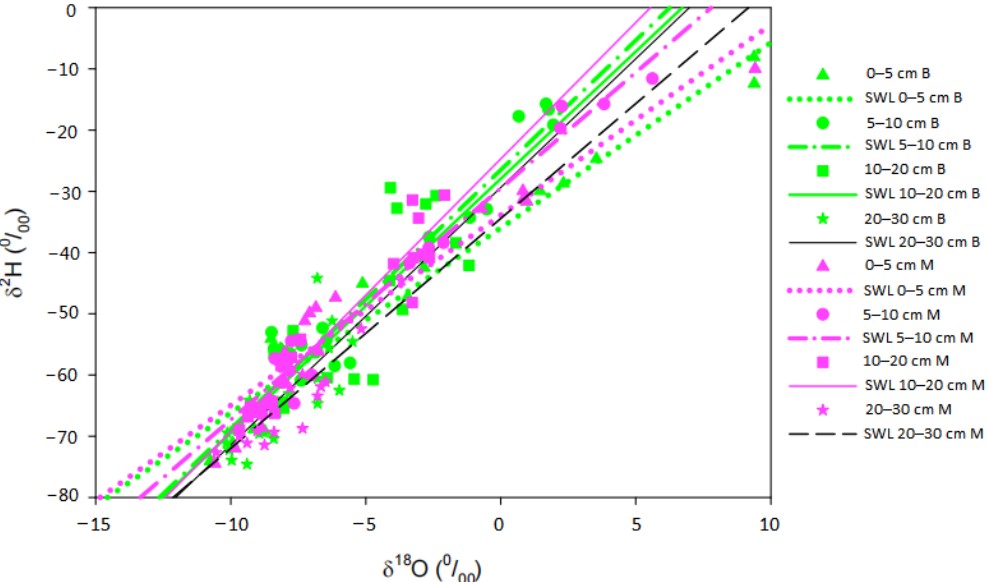

**Figure A2.** δ²H-δ¹⁸O relationship in soil water at different depth. *Bromus inermis* grassland (B) and *Medicago sativa* grassland (M).

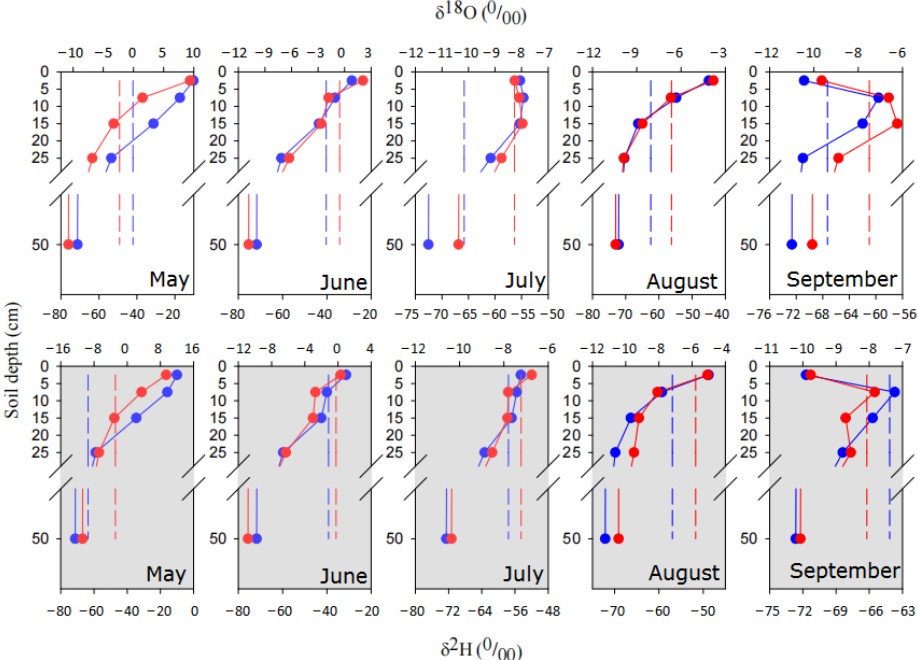

**Figure A3.** Intersection of δ[18]O and δ[2]H between soil water, groundwater and xylem water. δ[18]O and δ[2]H of soil water (0–30 cm), groundwater (50 cm) and xylem water in different months of *Bromus inermis* (white area) and *Medicago sativa* (gray area) grasslands. Red and blue symbols represent δ[18]O and δ[2]H, respectively. Red and blue dash lines denote δ[18]O and δ[2]H in xylem water, respectively.

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
