# Peer review of "Comparing Water Use Characteristics of Bromus inermis and Medicago sativa Revegetating Degraded Land in Agro-Pasture Ecotone in North China"

_water, doi:10.3390/w15010055_

Round 1

Reviewer 1 Report

Zhuo Pang, Hengkang Xu, Chao Chen, Guofang Zhang, Xifeng Fan, Juying Wu, Haiming Kan. Comparing water use characteristics of Bromus inermis and Medicago sativa revegetating degraded land in agro-pasture ecotone in North China

The relevance of this study is due to the fact that due to historical anthropogenic disturbances and recent climate change, APENC is environmentally vulnerable and is experiencing increasingly severe land degradation, water shortages, reduced productivity and loss of biodiversity. And the impact of revegetation by herbaceous plants on water conditions requires further study, and it is still unclear whether pastures will improve water loss or exacerbate the dryness of the soil layer. Regenerating plant species here include the native Bromus inermis and the exotic Medicago sativa. Therefore, Zhuo Pang et al. used the δ2H and δ18O stable isotopes to determine the sources of water uptake by these plants.

As a result of the studies performed, the authors conclude that in order to realize sustainable water resources utilization and ecosystem water balance, mixed planting Bromus inermis and Medicago sativa would be more desirable for revegetation of degraded lands in areas with both strong precipitation events and seasonal water shortage.

Some issues need to be clarified before publication.

2.2. Sample collection

A more detailed description of the sampling methodology is required. You write that you took 8 samples per month for 5 months. Should be 40 samples. However, Table 2 shows 14-19 samples.

Need a sampling scheme in the form of a picture.

The characteristics of the well from which water samples were taken for analysis should be given in more detail. What is its depth, for what purposes it is used, how stable water composition is during the year, what it depends on, and so on. You also need to explain why you consider the water from this well to be underground water. This requires a brief description of the lithology of the aquifer in relation to the soil horizons. I had such questions when I saw Figure A1. In my opinion, showing "groundwater line (GWL)" in this way is not correct. One sampling point is not enough to build a groundwater line. A deeper analysis of the composition of the sampled groundwater and the contribution of the results of studies of the isotopic composition of groundwater in this region by other researchers are needed. Figure A1 in your article is (in my opinion) evidence of the mixed composition of water samples that you consider to be underground. It seems that soil water is involved in its composition in some samples. GWL is usually close to LMWL.

Author Response

Zhuo Pang, Hengkang Xu, Chao Chen, Guofang Zhang, Xifeng Fan, Juying Wu, Haiming Kan. Comparing water use characteristics of Bromus inermis and Medicago sativa revegetating degraded land in agro-pasture ecotone in North China The relevance of this study is due to the fact that due to historical anthropogenic disturbances and recent climate change, APENC is environmentally vulnerable and is experiencing increasingly severe land degradation, water shortages, reduced productivity and loss of biodiversity. And the impact of revegetation by herbaceous plants on water conditions requires further study, and it is still unclear whether pastures will improve water loss or exacerbate the dryness of the soil layer. Regenerating plant species here include the native Bromus inermis and the exotic Medicago sativa. Therefore, Zhuo Pang et al. used the δ2H and δ18O stable isotopes to determine the sources of water uptake by these plants. As a result of the studies performed, the authors conclude that in order to realize sustainable water resources utilization and ecosystem water balance, mixed planting Bromus inermis and Medicago sativa would be more desirable for revegetation of degraded lands in areas with both strong precipitation events and seasonal water shortage. Some issues need to be clarified before publication. 2.2. Sample collection “A more detailed description of the sampling methodology is required. You write that you took 8 samples per month for 5 months. Should be 40 samples. However, Table 2 shows 14-19 samples.” Response: More detailed description of plant xylem and soil sampling was given in the revised manuscript (page 3 line 113 to page 4 line141 ). Specifically, “Four samples of plant xylem and soil at different depths were collected in subplots per month from May to September. There should be 20 xylem samples for each grassland type and 20 soil samples for each soil layer.” However, “ If the sample was too dry to extract enough water for measurement, then extracted water of 2 replicating samples would be merged as one sample. So the number of xylem and soil water samples might be less than 20.” “Need a sampling scheme in the form of a picture.” Response: A sampling scheme was provided in the revised manuscript as Figure 2 (page 3 line 109 to 111), to display the experimental plots, soil horizons and aquifer. “The characteristics of the well from which water samples were taken for analysis should be given in more detail. What is its depth, for what purposes it is used, how stable water composition is during the year, what it depends on, and so on. You also need to explain why you consider the water from this well to be underground water. This requires a brief description of the lithology of the aquifer in relation to the soil horizons. “ Response: The characteristics of the well and the relation between the aquifer and soil horizons were given in the revised manuscript (page 3 line 117 to page 4 line 119 AND page 4 line 154 to page 4 line 157). Specifically, “The sampling well, with a depth of 2 m, was used for water quality monitoring and the aquifer beneath the soil horizons is pebbly sand.” “The stable isotopic composition of groundwater depends on the recharging river water and has little change during the year. The groundwater in aquifer can influence the stable isotopic composition of soil water in the bottom layer through capillary action.” “I had such questions when I saw Figure A1. In my opinion, showing "groundwater line (GWL)" in this way is not correct. One sampling point is not enough to build a groundwater line. A deeper analysis of the composition of the sampled groundwater and the contribution of the results of studies of the isotopic composition of groundwater in this region by other researchers are needed. Figure A1 in your article is (in my opinion) evidence of the mixed composition of water samples that you consider to be underground. It seems that soil water is involved in its composition in some samples. GWL is usually close to LMWL.” Response: Thanks for the suggestion, the groundwater line (GWL) has been removed as suggested in the revised manuscript (page 9 line 244 AND Figure A1). The mixed isotopic composition between soil water and groundwater has been further analyzed (page 15 line 485 to page 15 line 494). Specifically, “Additionally, the sampled groundwater is recharged by the nearby Gui River and its isotopic composition (Table 2) falls within the range of groundwater in Beijing area (-92‰ to -52‰ for δ2H and -13.2‰ to -6‰ for δ18O) [75]. The mixed isotopic composition between groundwater and 0-5 cm soil water in September may be caused by a precipitation event, in that the rainwater, with a stable isotopic composition (-74.87‰ for δ2H and -11.98‰ δ18O, Figure A1) similar to groundwater (Table 2), replenished 0-5 cm soil layer. However, the contribution of 0-5 cm soil water and groundwater to Bromus inermis and Medicago sativa water consumption in September was small (Figure 7), so the mixed isotopic composition did not exert significant influence on water uptake source analysis.”

Reviewer 2 Report

Authors attempted to quantify the contributions of various water sources to plant water uptake and water use characteristics and influences on soil water dynamics between Bromus inermis and Medicago sativa with the use of stable isotopes. Its quite interesting subject, and serve as an advanced reference in crop water relation studies. The article is well written. However, it needs to adress the following issues:

1. In the methodology - more explanation is needed - how authors implemented stable isotopes treatments to studied plants. 

Author Response

Authors attempted to quantify the contributions of various water sources to plant water uptake and water use characteristics and influences on soil water dynamics between Bromus inermis and Medicago sativa with the use of stable isotopes. Its quite interesting subject, and serve as an advanced reference in crop water relation studies. The article is well written. However, it needs to adress the following issues: “1. In the methodology - more explanation is needed - how authors implemented stable isotopes treatments to studied plants” Response: More explanation was provided in the methodology about stable isotopes treatments to studied plants. (page 4 line 152 to page 4 line 158) Specifically, “Both soil water and groundwater are primary plant water sources on the floodplain because plant roots can reach the groundwater due to its shallow buried depth. The stable isotopic composition of soil water depends on depth and becomes more enriched up to the surface layer due to the fractionation effect of evaporation. The stable isotopic composition of groundwater depends on recharging source and has little change during the year. The groundwater in aquifer can influence the stable isotopic composition of soil water in the bottom layer through capillary action. Xylem water is a mixture of uptake sources depending on the root characteristics of plant types.”

Round 2

Reviewer 1 Report

The authors have addressed all my comments to great extent

 I recommend paper publication
